# Cryo-EM structure of cyanophage P-SCSP1u offers insights into DNA gating and evolution of T7-like viruses

Lanlan Cai [1,2,3,7], Hang Liu [4,7], Wen Zhang[4], Shiwei Xiao[1], Qinglu Zeng [1,3,5] ✉ & Shangyu Dang [4,6] ✉

Cyanophages, together with their host cyanobacteria, play important roles in marine biogeochemical cycles and control of marine food webs. The recently identified MPP-C (Marine Picocyanobacteria Podovirus clade C) cyanophages, belonging to the T7-like podoviruses, contain the smallest genomes among cyanopodoviruses and exhibit distinct infection kinetics. However, understanding of the MPP-C cyanophage infection process is hindered by the lack of high-resolution structural information. Here, we report the cryo-EM structure of the cyanophage P-SCSP1u, a representative member of the MPP-C phages, in its native form at near-atomic resolution, which reveals the assembly mechanism of the capsid and molecular interaction of the portal-tail complex. Structural comparison of the capsid proteins of P-SCSP1u and other podoviruses with known structures provides insights into the evolution of T7-like viruses. Furthermore, our study provides the near-atomic resolution structure of portal-tail complex for T7-like viruses. On the basis of previously reported structures of phage T7, we identify an additional valve and gate to explain the DNA gating mechanism for the T7-like viruses.

Marine picocyanobacteria, with only two genera, *Prochlorococcus* and *Synechococcus*, numerically dominate the marine phytoplankton and contribute up to 50% of primary production in marine ecosystems[1–5]. Cyanophages (viruses that infect cyanobacteria) are extremely abundant and widely distributed in the world's oceans, where they play important roles in regulating the metabolic activity, evolution, and community structures of cyanobacteria[6–9]. The infection and lysis of cyanobacteria by cyanophages significantly influence the global biogeochemical cycles[10,11]. For example, cyanophages can modify host photosynthesis by inhibiting $CO_2$ fixation to maximize energy for phage propagation, leading to an estimated loss of 0.02–5.39 petagrams of fixed carbon per year[12].

Marine cyanophages belonging to the class *Caudoviricetes* consist of three morphologically and genetically distinct groups: T7-like cyanopodoviruses with short tails[13–15], T4-like or S-TIM5-like cyanomyoviruses with long contractile tails[16–19], and cyanosiphoviruses with long non-contractile tails[20,21]. Although sharing similar genomic architecture with the *Escherichia coli* phage T7, the T7-like cyanopodoviruses differ from phage T7 in several genes, some of which have been verified to encode proteins contributing to the stabilization of virion, the adsorption onto host cells, as well as the subsequent reprogramming of host cellular metabolism[22–24]. Based on genomic analyses, T7-like cyanopodoviruses have been further divided into three discrete clades: MPP (Marine Picocyanobacteria Podovirus) -A, -B, and -C

[1]Department of Ocean Science, The Hong Kong University of Science and Technology, Clear Water Bay, Hong Kong, China. [2]Southern Marine Science and Engineering Guangdong Laboratory (Zhuhai), Zhuhai, China. [3]HKUST Shenzhen-Hong Kong Collaborative Innovation Research Institute, Shenzhen, China. [4]Division of Life Science, The Hong Kong University of Science and Technology, Clear Water Bay, Hong Kong, China. [5]Center for Ocean Research in Hong Kong and Macau, The Hong Kong University of Science and Technology, Hong Kong, China. [6]HKUST-Shenzhen Research Institute, Nanshan, Shenzhen 518057, China. [7]These authors contributed equally: Lanlan Cai, Hang Liu. ✉e-mail: zeng@ust.hk; sdang@ust.hk

clusters[13–15,25,26]. Virus particle structures have been solved to reveal the structures and infection processes of the MPP-A and MPP-B phages[23,24,27], although with incomplete information due to technology limitations. However, we still know little about the structures of MPP-C phages and their infection mechanisms.

Technology breakthroughs in single-particle cryo-electron microscopy (cryo-EM) enabled atomic resolution structural determination of the virion (i.e., an infectious virus particle) to understand the molecular mechanisms of virion assembly and infection of their hosts[28]. *E. coli* phage T7, with an icosahedral capsid and a short tail, is a model virus for understanding the assembly, infection process, and DNA packaging mechanisms common to tailed dsDNA phages and herpesviruses[29–31]. The structures of mature T7 phage capsid[29,31], DNA-free capsid[31], isolated portal, and fiberless tail complex[32] have been determined at medium to near-atomic resolutions. As the currently known simplest phage model, the structure of T7-like phages, consisting of a short tail attached to a pentameric vertex of the icosahedral capsid shell through a dodecameric portal[29], has the advantage of comprehensive and detailed comparison between different members within the group. So far, only two marine cyanophages, Syn5 and P-SSP7 (both are T7-like podoviruses), have been structurally determined, mainly focusing on capsid proteins. The *Synechococcus* phage Syn5, an MPP-A virus, carries knob-like proteins in the icosahedral capsid, along with an unusual horn-like structure on the opposite vertex of the tail to stabilize the capsid[24]. Unlike Syn5, the *Prochlorococcus* phage P-SSP7 belongs to the MPP-B clade and solely consists of major capsid proteins in the capsid shell, suggesting a different mechanism in the capsid assembly[23]. Compared to MPP-A and MPP-B phages, the recently reported MPP-C phages show the smallest genomes and longest infection periods[15]. To date, all the MPP-C phages were isolated from *Prochlorococcus* MED4 and abundantly distributed in the global oceans[15]. However, understanding the assembly and infection mechanisms of MPP-C phages is hindered by the lack of high-resolution structures, particularly the portal-tail complex that plays a crucial role in virus infection.

To expand our understanding of the diversity and evolution of T7-like cyanophages and to provide more information for the phage infection process, we determine the high-resolution structure of P-SCSP1u, a representative member from the MPP-C clade, in its native form by single-particle cryo-EM. Structural comparison with phages from the MPP-A and MPP-B clades, as well as the T7 phage, suggests a structure-based classification of T7-like viruses. Together, our results provide in-depth knowledge of the structure of a cyanopodovirus at the molecular level and offer insights into the infection process and evolution of T7-like viruses.

## Results

### Overview of the P-SCSP1u virion

The MPP-C phage P-SCSP1u was harvested after infecting the host strain *Prochlorococcus* MED4 and purified by cesium chloride density gradient ultracentrifugation to a concentration of ~10[13] virus particles/mL. The purity and homogeneity of the sample were checked by negative stain electron microscopy. By using single-particle cryo-EM, we finally determined the structure of P-SCSP1u in its native state at 3.2 Å and 3.8 Å for the capsid and portal-tail complex, respectively (Fig. 1, Supplementary Figs. 1 and 2, Supplementary Table 1). Overall, the P-SCSP1u virion possesses an icosahedral capsid with a diameter of ~600 Å and a short tail of ~190 Å in length (Figs. 1a, 2a). Compared to Syn5 (660 Å, the MPP-A clade) and P-SSP7 (655 Å, the MPP-B clade)[23,24], P-SCSP1u has the smallest capsid, consistent with its smallest genome size among reported cyanopodoviruses[14,15,20]. In the interior of the P-SCSP1u capsid, which is ~20 to 30 Å thick, the concentrically

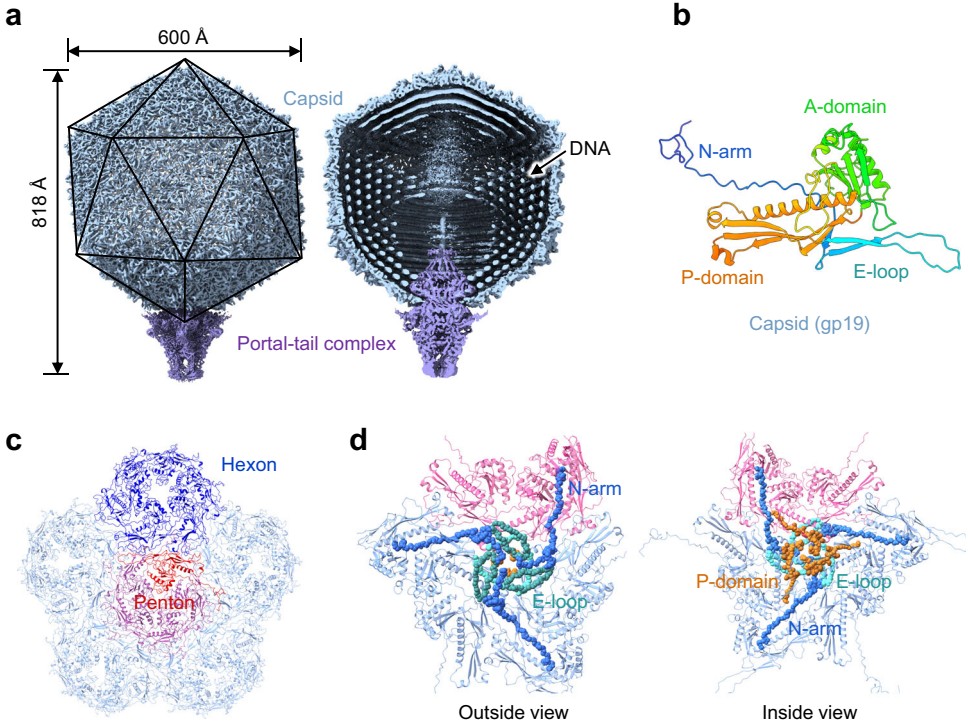

**Fig. 1 | The overall architecture of the P-SCSP1u virion. a** Surface (left) and section (right) views of the P-SCSP1u virion. Packed DNA inside the capsid is indicated by an arrow and the portal-tail complex is shown in purple. The sizes of the intact virion are labeled. **b** Structure of the major capsid protein (gp19) with domains colored differently. **c** Each vertex of the P-SCSP1u capsid includes one central penton (red) and five surrounding hexons (blue). An asymmetric unit (T = 7) consisting of one hexon (dark blue) and one subunit of penton (dark red) is highlighted. **d** The interactions of the N-arm, E-loop, and P-domain of the adjacent subunits from three capsomers (one penton and two hexons), viewed from outside (left) and inside the capsid (right). The backbone of the N-arm, E-loop, and P-domain are shown as spheres with van der Waals radii.

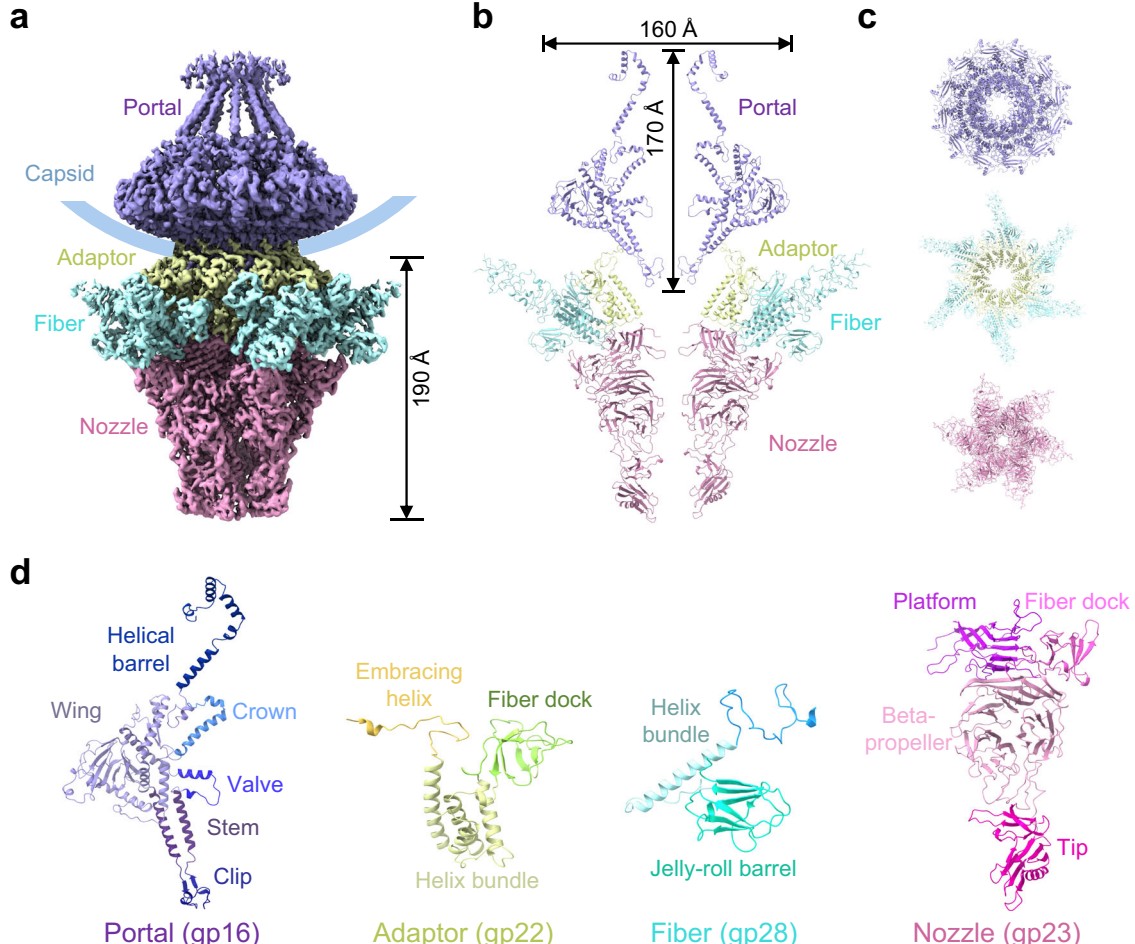

**Fig. 2 | Assembly of the P-SCSP1u portal-tail complex. a** Surface view of the portal-tail complex with components colored differently: purple for the portal, yellow for the adaptor, cyan for the fiber N-termini, and magenta for the nozzle. The colors are as in (**b**) and (**c**). The size of the tail machine is labeled. The position of the capsid is indicated as a blue curve. **b** Two opposing chains of different components in the portal-tail complex. The sizes of the portal are labeled. **c** Top view of the portal (purple), adaptor (yellow), fiber N-termini (cyan), and nozzle (magenta) of the P-SCSP1u. **d** Structure of different components in the portal-tail complex with domains colored differently.

packaged DNA is arranged in multiple layers with a ~25-Å space between two layers (Fig. 1a). A cylinder-shaped portal was observed to attach to the inner surface at one pentameric vertex of the capsid shell and extended outside of the capsid to connect with the adaptor (Figs. 1a, 2a). The region of the portal protein inside the capsid is surrounded by DNA (Fig. 1a), which is commonly observed in the tailed phages[23,29]. The outer tip of the portal protein interacts with the ring-like adaptor by the subsequent incorporation of the tubular nozzle, forming a DNA ejection channel (Fig. 2, see below).

### The capsid structure of P-SCSP1u

By using sub-particle and symmetry expansion in the data processing[33], the resolution of capsid protein (gp19, 328 aa) has been improved to 3.2 Å (Supplementary Figs. 1 and 2, Supplementary Table 1), enabling the de novo model building covering nearly all residues (Ala2 to Ser328) unambiguously (Fig. 1b). The high resolution of P-SCSP1u enabled us to confidently elucidate the fine-scale secondary structural elements of the protein subunits, such as long loops and large β-sheets. Similar to T7 and the MPP-B cyanophage P-SSP7, P-SCSP1u does not have decoration (cement) proteins in the icosahedral capsid. The whole capsid of P-SCSP1u is solely composed of 415 copies of major capsid protein (gp19), exhibiting 60 hexameric capsomeres (hexons) and 11 pentameric capsomeres (pentons). Conversely, the MPP-A cyanophage Syn5 carries knob-like and horn-like

proteins in the capsid, which were proposed to contribute to capsid stabilization[24]. Each vertex of the P-SCSP1u icosahedral head includes one central penton and five surrounding hexons, which can be divided into five asymmetric units (Fig. 1c). As observed in T7 and two cyano-podoviruses (P-SSP7 and Syn5), each asymmetric unit of the P-SCSP1u capsid consists of one hexon and one subunit of penton, yielding a triangulation (T) number of 7 for P-SCSP1u.

Each capsid protein of P-SCSP1u contains four distinct domains from N to C terminus (Fig. 1b): an extended N terminus (N-arm), a long-extended loop (E-loop), a triangular domain (A-domain) and an elongated protrusion domain (P-domain). The N-arm, E-loop, and A-domain are mainly formed by the highly flexible long loops, which enable easy conformational adjustment to connect with neighboring capsid proteins for assembly. Although sharing low sequence similarity with the major capsid protein (gp5, 385 aa) of the bacteriophage HK97, the P-SCSP1u capsid protein gp19 presents the HK97-like fold[34], which was first determined in phage HK97[35] and has been discovered in many tailed phages and herpesviruses[34,36]. This observation suggests the evolutional connection between P-SCSP1u and viruses with the HK97-like fold.

To reveal how the capsid proteins interact to maintain the stability of the capsid, we identify the detailed interactions within and between capsomers of the P-SCSP1u capsid. The pentons and hexons are assembled by similar intra-capsomeric interactions, with A-domains

constituting the core of capsomers and E-loops wrapping around the adjacent N-arms and P-domains, forming a circulating interaction mode to stabilize the capsomer (Fig. 1c, Supplementary Fig. 3a–c). The inter-capsomeric interactions are mediated by mutual penetration of the E-loops and P-domains of neighboring capsomers and further rein-forced by the extended N-arms interacting with multiple subunits to fasten the adjacent capsomers (Fig. 1d, Supplementary Fig. 3d–f). PISA analyses further revealed substantial hydrogen bonds formed among subunits to provide strong support for the assembly of the capsid. In detail, the A-domain and E-loop of the penton interact with the N-arm of the hexon, and the P-domains of two subunits form hydrogen bonds with each other (Supplementary Fig. 3d, e). For hexon-hexon interac-tions, two interfaces are generated by four subunits. The N-arm of one hexon binds with the P-domain and E-loop of two subunits of the other hexon in every interface (Supplementary Fig. 3f). Interestingly, com-pared with the pentons, the subunits of hexons form slightly strong interactions in the capsomer with more hydrogen bonds because of the different conformational curvature in the N-arm and E-loop regions (Supplementary Fig. 3). The N-arm, E-loop, and P-domain of three subunits from three adjacent capsomers interact with each other with more than 30 hydrogen bonds to tightly connect the capsomer. Overall, the capsid of P-SCSP1u is assembled via non-covalent interactions to maintain the high internal pressure for DNA packaging.

## Assembly of the P-SCSP1u portal-tail complex

The portal-tail complex of P-SCSP1u consists of dodecameric portal protein (gp16), dodecameric adaptor protein (gp22), hexameric nozzle protein (gp23), and hexameric trimer of fiber proteins (gp28) (Fig. 2), in a similar organization to those of T7[29,32] and the MPP-B cyanophage P-SSP7[23]. By using the symmetry-mismatch reconstruction method[33], the asymmetrical portal-tail complex of P-SCSP1u was reconstructed to 3.8 Å resolution (Supplementary Figs. 1 and 2, Supplementary Table 1), enabling us to study the detailed assembly between each subunit.

The cylinder-shaped P-SCSP1u portal, 170 Å in height with the largest external diameter of 160 Å, is formed by 12 copies of gp16 that are arranged around an axial central channel (Fig. 2a–c). Similar to the portal protein of T7 (gp8)[29], the portal protein of P-SCSP1u is com-posed mainly of α-helices and can be divided into five domains: C-terminal helical barrel, crown, wing, stem, and clip (Fig. 2d)[29,32]. The valve, a succeeding loop connecting the wing and the stem in the portal protein, points perpendicularly to the channel (Fig. 2d), which was predicted to play essential roles in regulating DNA entry and exit[32]. The clip regions of the portal provide a basis for connecting 12 copies of the adaptor protein gp22, which bridge all other components in the portal-tail complex (Fig. 2b, d). The adaptor protein (gp22) is com-posed of five α-helices and five β-strands and is divided into three domains (Fig. 2d): a C-terminal embracing helix, an up-down α-helix bundle, and a fiber dock. Six subunits of the gp23 nozzle protein form a hexamer that attaches to the helical bundle region of the adaptor (Fig. 2b, c). The nozzle monomer can be divided into four domains: the platform, the fiber dock, the large central β-propeller domain, and the tip domain at the most distal part (Fig. 2d).

The tail fiber of T7 is supposed to comprise two domains, the N-termini binding to the adaptor-nozzle complex and the C-termini (distal part) interacting with the host receptor during infection[37]. Each of the six fiber trimers of P-SCSP1u (gp28) is fitted between the nozzle fiber dock and the adaptor fiber dock. The N-termini of tail fiber, anchoring to the adaptor-nozzle fiber docks, was clearly resolved in our structure (Fig. 2, Supplementary Figs. 1 and 2, Supplementary Table 1). Each fiber N-terminus is formed by a jelly-roll barrel and an α-helix (Fig. 2d). However, the distal parts of the fibers could not be resolved in the cryo-EM density map, suggesting the relatively flexible assembly of these components without attaching to the host. Con-sistently, so far, the in situ high-resolution structure of intact tail fiber has not been determined for tailed phages.

The symmetry mismatch between the five-fold capsid vertex and the 12-fold portal-adaptor complex is a characteristic of tailed bacteriophages[38]. To reveal more details about the capsid and portal-tail mismatch, we reconstructed the cryo-EM density map of the portal-tail complex without symmetry at 4.6 Å (Supplementary Figs. 1 and 4). The result showed that the portal-tail complex replaces one penton of the icosahedral capsid and is clipped by five hexons (namely mis-matched hexons) in a manner analogous to the pentons in other ver-tices. Structural comparison of capsid proteins between mismatched hexons and other vertices revealed only slight conformational changes in N-arm and P-domain regions of the capsid proteins (Supplementary Fig. 5a). However, various interactions occur between the capsid and the portal/adaptor in the portal vertex (i.e., the unique five-fold vertex where the dodecameric portal protein replaces one penton of the capsid). The N-arm, P-domain, and E-loop of the capsid subunits from the five mismatched hexons form a ring-like structure and are clipped by the circular groove of portals and adaptors (Supplementary Fig. 4c, d). The large wing regions of the dodecameric portal form a wide surface for the capsid to bind from inside, while the embracing helixes of the adaptor provide a smaller surface to stabilize the central part of the capsid shell (Supplementary Fig. 4d, e).

## Interaction between the portal and the tail components

The high-resolution structure of the portal-tail complex of P-SCSP1u enabled us to investigate the multiple interactions among different components at the molecular level. Briefly, five interfaces were iden-tified to play crucial roles in strengthening the assembly of the portal-tail complex of P-SCSP1u (Fig. 3). The clip regions of two neighboring portal subunits (named P1 and P2) interact with the embracing helix and the helix bundle of one adaptor subunit (named A2) (Fig. 3a). An approximate 500 Å² interface area formed by 10 residues of P1 and 15 residues of A2 was identified based on PISA interface analysis[39]. Fur-thermore, 22 residues of P2 and 24 residues of A2 form an interface area of around 700 Å². The E168, H176, and M178 of interacting adaptor subunit (A2) form hydrogen bonds with T312, I321, and G319 of the P1 clip region. The S177 (the embracing helix) and E50 (the helix bundle) of the A2 adaptor interact with the S314 and R309 from the P2 clip region by forming hydrogen bonds and salt bridge respectively to further strengthen the interaction (Fig. 3b). For the interface between the adaptors and nozzles (Fig. 3c), a 500 Å² area is formed by 16 resi-dues of one nozzle subunit (named N1) and 18 residues of one adaptor subunit (named A3). Another 14 residues of N1 interact with 15 residues of another adaptor subunit (named A2), forming a 420 Å² interface. The platform region of N1 tightly interacts with the helix bundle regions of A2 and A3 by two pairs of hydrogen bonds (R766-E20 and D737-N24) and one pair of salt bridges (R766-E19) (Fig. 3a, c). Collec-tively, the dodecameric portal, dodecameric adaptor, and the hex-americ nozzle in the portal-tail complex are intimately connected with each other, with a total of 60 hydrogen bonds and 18 salt bridges, providing a stable platform for DNA ejection.

The N-terminus domains of three fiber subunits form a homo-trimer as a functional unit via simultaneous binding to the adaptor and the nozzle. We name these three fiber subunits F1, F2, and F3 clockwise according to the side view of the tail complex for easy description (Fig. 3a). The interlaced protein chains produce complicated interac-tions among adaptor–nozzle–fiber (Fig. 3), strengthening the structure binding. The helical bundles of three fiber monomers (F1, F2, F3) form around 15 hydrogen bonds with residues from the helical bundles of two adaptor monomers (A1, A2) and the platform of one nozzle (N1) (Fig. 3d). These observations suggested that the two adaptors and one nozzle may form a stable foundation for the possible rotation of fiber during the infection process (see discussion below). The jelly-roll barrel regions of two of three fibers (F1 and F2) bind with the fiber dock regions of two adaptors (A1 and A2), respectively (Fig. 3e, f). The binding of F1 to the adaptor involves more hydrogen bonds than that

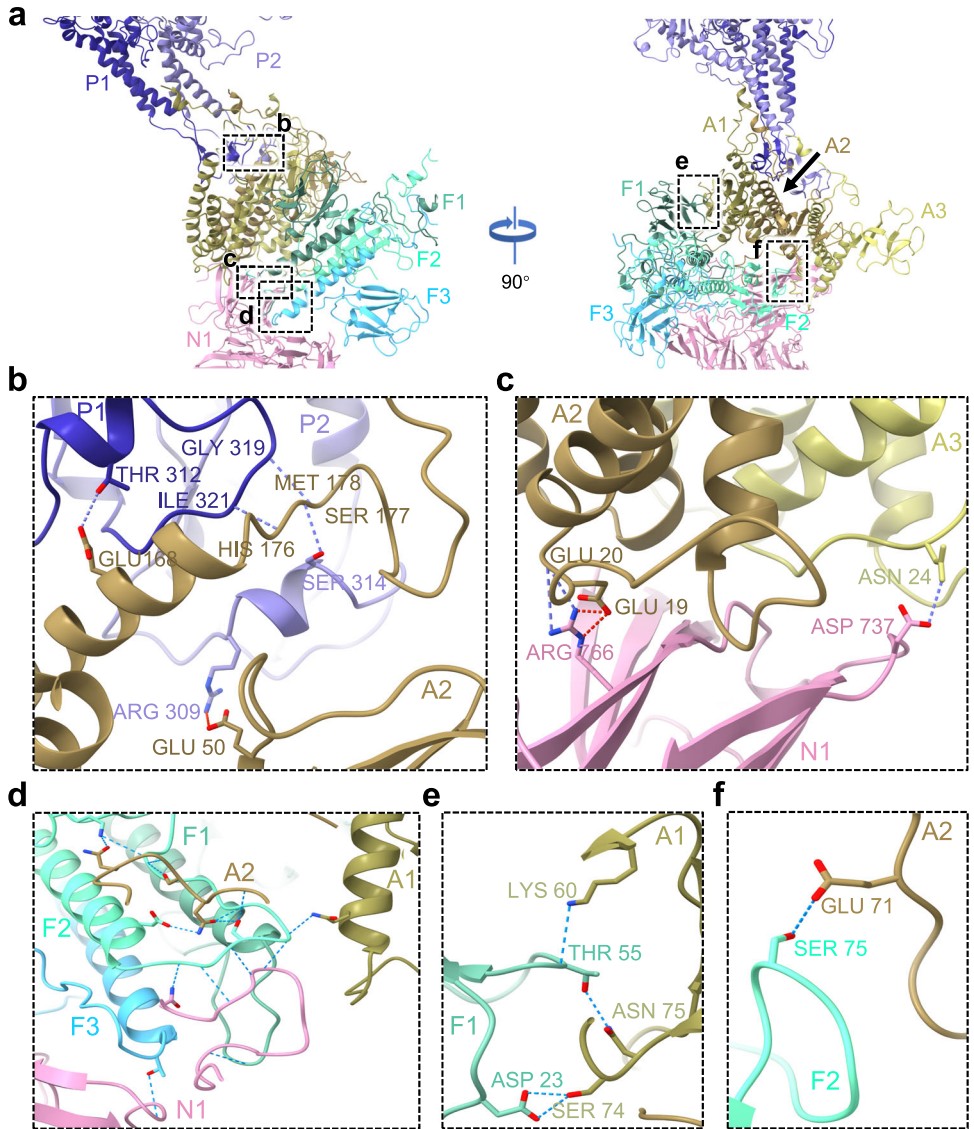

**Fig. 3 | Interactions among different components in the P-SCSP1u portal-tail complex. a** The side views of portal-tail components are shown in two directions. Two portal subunits (P1 and P2), two adaptor subunits (A1 and A2), three fiber subunits (F1, F2, and F3), and one nozzle subunit (N1) are shown and colored differently. The colors are as in Fig. 2. Interaction details of five interfaces highlighted with dashed boxes are shown in higher magnification in the following panels (**b–f**). The residues responsible for interactions are shown as sticks and labeled. The hydrogen bonds and salt bridges are indicated as blue and red dashed lines, respectively.

of F2, which may lead to the tilting of the fiber homotrimer in the horizontal direction from the top view (Supplementary Fig. 2i). In contrast, F3 contributes the least to the binding of fibers onto the nozzles, and no hydrogen bond is found between the jelly-roll barrel region of F3 and the nozzle, which may be the reason for the upward tilting of fibers against the tail complex (Fig. 3a, d, Supplementary Fig. 2i). In the fiber docking region of the nozzle, there is no evident interaction with F2 observed though they are close to each other. In conclusion, we find that every monomer of the fiber homotrimer owns different binding interfaces and interaction force levels, which may result in the specific angle of the fiber against the axis of the tail complex.

**DNA ejection channel formed by portal-tail complex**
The portal, adaptor, and nozzle proteins form a channel for DNA ejection (Fig. 4). A columnar density, which could be assigned to the packaged DNA in the capsid[29,40], passes through the central channel of the portal and ends at the portal-nozzle interface in P-SCSP1u capsid

(Fig. 4b). Structural analysis revealed the fluctuation of the radii of the channel along the portal-tail complex (Fig. 4a). The various electrostatic potential distribution of the inner surface of the DNA channel also suggests the diverse function of the regions along the channel (Supplementary Fig. 2g).

The succeeding loop of the dodecameric portal protein constitutes one of the narrowest sections of the channel (Fig. 4a), which was previously identified as a "valve" and suggested to function as clamps to retain packaged DNA in the phage T7[32]. Structural analyses showed that the valve of the mature P-SCSP1u portal presents a similar shape and angle to that of the mature T7 (middle panel in Fig. 4d). The peptide chains with potential interactions with DNA in the valve of the P-SCSP1u capsid contain mainly positively charged amino acids (Supplementary Fig. 2g). These positively charged amino acids could enable electrostatic interaction with the negatively charged DNA, therefore stabilizing and protecting DNA from escaping from the mature capsid. The obvious conformational changes of the valve in T7 before and after DNA injection also prove its critical role in DNA

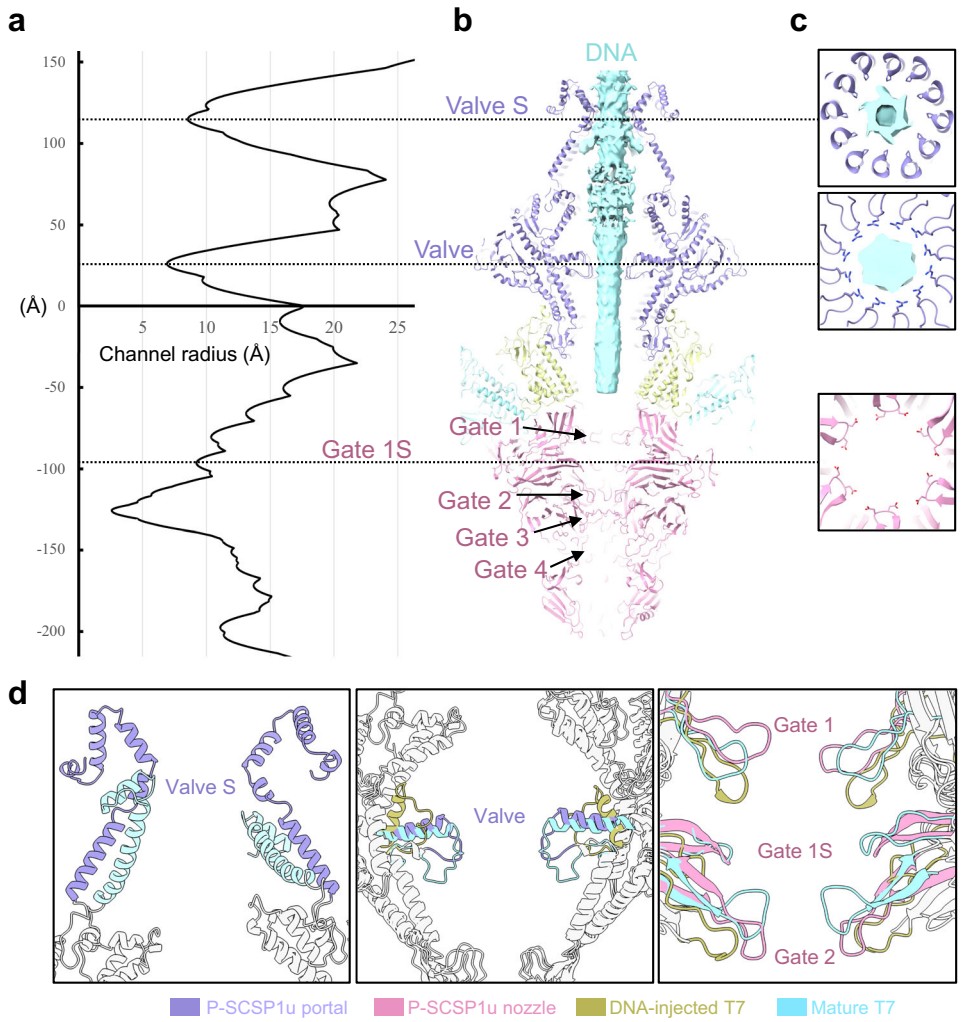

**Fig. 4 | Analyses of DNA channel in the portal-tail complex. a** The radius curve of the DNA channel along the portal-tail complex. **b** Section view of the portal-tail complex with four components colored differently. The colors are as in Fig. 3. The cryo-EM density corresponding to putative DNA along the center of the channel is indicated as a cyan cylinder. Valves and gates are indicated. **c** Cross-section views of three narrow parts (including Valve, Valve S, and Gate 1S) of the DNA ejection channel. The side chains of the key residues are shown as sticks. **d** Structural comparison of portal-tail complex among the mature P-SCSP1u (purple for portal and magenta for nozzle), the mature T7 (cyan, PDB:7EY8), and DNA-injected T7 (yellow, PDB:7EY6). Differences in valves and gates are highlighted and labeled. The Valve S of DNA-injected T7 is invisible in the cryo-EM structure.

retention[29,32]. According to the radii of the channels, an additional narrow "valve", which we named valve S, is identified in the P-SCSP1u capsid (Fig. 4). Valve S, with a radius less than 9 Å, is surrounded by 12 prolines from the helical barrel of the dodecameric portal proteins (Fig. 4c). The resistance between the hydrophobic proline ring of valve S and the hydrophilic DNA may help fix DNA in the center of the channel. Meanwhile, the helical barrel region, including valve S, owns substantial flexibility and can not be seen in structural models of the DNA-injected T7[29,32], which indicates that the interaction between DNA and valve S might also stabilize the structure of the portal barrel. Furthermore, compared with the structure of the portal protein in mature T7[29], the P-SCSP1u portal with a longer helical barrel region forms a longer and larger pore channel (left panel in Fig. 4d). These structural observations suggest the potentially critical role of the helical barrel of the P-SCSP1u portal in the DNA gating system of P-SCSP1u.

Four gates (Gates 1–4), formed by loops in nozzles, have been identified previously from the assembled tail complex of T7[32]. Similarly, those four gates were observed in the nozzles of P-SCSP1u (Fig. 4b). Gate 1, located at the top of the nozzle, has a radius larger than 10 Å (Fig. 4a, b). Furthermore, gate 2 and gate 3 form a pore

with a radius smaller than 3 Å. The radius of the pore formed by gate 4 is larger than 10 Å. Intriguingly, an additional gate, which we named gate 1S, is identified in the middle of the nozzle located between gates 1 and 2 (Fig. 4b). Although gate 1S possesses a pore with a radius larger than 9 Å, it is formed by the negatively charged aspartic residues in the loops of the nozzle platforms, different from the neutrally charged residues in gate 1 (Fig. 4c, Supplementary Fig. 2g). These results indicate that gate 1S may play a role in the DNA gating system of the portal-tail complex. A potential mechanism to protect DNA from escaping the capsid is that the steric resistance of the narrow channel in gates 2 and 3 forms a strong gating system, while gate 1S provides additional electrostatic repulsion force. These results show that the blocking effect of nozzles on DNA is possibly formed through different forces from multiple regions. Despite the differences in the loop shapes, all gates of the P-SCSP1u nozzle have similar positions and directions as those of mature T7[29], suggesting a conserved function of nozzles in DNA ejection among T7-like phages (right panel in Fig. 4d). In addition, all gates, including the newly identified gate 1S, of the DNA-injected T7 nozzle are in the open state (right panel in Fig. 4d), further demonstrating their critical roles in DNA securing.

## Discussion

We determined the high-resolution cryo-EM structure of P-SCSP1u in its native state, providing detailed information to understand the assembly of both capsid shell (3.2 Å) and portal-tail complex (3.8 Å) for P-SCSP1u, a representative member of MPP-C cyanophages. The overall component of the P-SCSP1u capsid shell is similar to that of phage T7 and the MPP-B cyanophage P-SSP7 but different from that of the MPP-A cyanophage Syn5. Previously, the portal-tail complex has been structurally determined for only one T7-like cyanophage (P-SSP7) at 9 Å resolution, hindering the precise assignment of different protein components in the portal-tail complex[23]. The dissection of the whole P-SCSP1u virion in our study gives a near-atomic view of the spatial organization of different protein components and the complexity of molecular interactions that accommodate the stabilization and infection of the phage particle. In addition, based on previous structural studies of T7, we identified an extra valve and a gate in the DNA channel of the portal-tail complex, refining our understanding of the DNA gating mechanism for T7-like phages.

Conventionally, sequence-based methods are used for viral phylogenetic analysis and classification, which forms the basis of the hierarchical taxonomic system governed by the International Committee on Taxonomy of Viruses. However, due to the complex relationship between gene sequences and protein functions, genetic information has limitations in understanding viral phylogeny and biology. Over the past decades, structural biology has undergone dramatic improvements, leading to a rapid growth in the number of viral protein structures and allowing for structure-based analysis of viral phylogeny[28,41]. We used the structure-based method for the analysis of the evolutionary relationship among T7-like viruses. We compared the capsid protein structures of T7 (gp10, 347 aa), MPP-A cyanophage Syn5 (gp39, 332 aa), MPP-B cyanophage P-SSP7 (gp10, 375 aa), and MPP-C cyanophage P-SCSP1u (gp19, 328 aa) since the structure information of portal-tail complex is absent for Syn5 and relatively limited for P-SSP7[23,24]. Pairwise sequence alignment shows that gp19 of P-SCSP1u shares 51.45% similarity with gp39 of Syn5, slightly higher than that with gp10 of P-SSP7 (50.39%) and gp10 of T7 (49.29%) (Supplementary Figs. 5 and 6), suggesting that the capsid of P-SCSP1u is phylogenetically closer to that of Syn5. To quantitatively investigate the structural similarity among different viruses, the Dali $Z$-scores of the capsid proteins were calculated[42]. The capsid of P-SCSP1u shares a higher structural similarity with that of T7 ($22 < Z$-score $< 25$) than Syn5 ($13 < Z$-score $< 15$) and P-SSP7 ($9 < Z$-score $< 11$). One of the major differences comes from the A-loop in the A-domain that is only found in T7-like phages so far. The length of this loop in P-SCSP1u is similar to that of T7, while is shorter than those of P-SSP7 and Syn5 (Supplementary Fig. 5, Supplementary Table 2). Consistently, the Dali $Z$-scores revealed that seven capsid protein subunits in the viral asymmetric unit of P-SCSP1u show higher similarities with those of T7 (Supplementary Fig. 6). These structural analyses suggest that the capsid of P-SCSP1u is structurally closer to that of T7, rather than two cyanopodoviruses Syn5 and P-SSP7. Indeed, a drawback of classification based solely on sequence is that proteins with high sequence similarity do not necessarily have high structural similarity. The results of our analysis provide a glimpse into the potential application and development of the structure-based classification of proteins. We propose that as more structures of proteins are solved, structure-based analysis, in combination with sequence-based analysis, has the increasing potential to optimize the classification of viruses at a holistic level, especially in the case that proteins share similar folds, but the similarity is perhaps too subtle to be detected by the sequence-based approach.

To complete the infection process of recognizing and binding to the host and subsequently injecting the packed DNA into the host cell, different components of the virion work together in a highly organized way. The flexibility of the tail fiber, particularly in the C-terminal domain, ensures high efficiency in interaction with host receptors. A previous study of T7 has observed conformational changes in the fiber N-terminal domain from the mature state to the DNA-injected state after interacting with the host receptor[40]. The N-terminal fiber of mature P-SCSP1u resembles that of mature T7, while the tilting angle is slightly larger (Supplementary Fig. 2i). Our high-resolution structure of the portal-tail complex exhibits the asymmetric interaction of three fiber monomers (F1, F2, and F3) and other tail parts (Fig. 3), providing information to understand the unique conformation of fiber for P-SCSP1u. During the DNA injection, binding of the C-terminal fiber to the host cell may trigger the conformational changes in the N-terminal fiber, switching it from an upward state to a horizontal state (Fig. 5)[30,40,43]. The horizontal state of the fiber could be stabilized by the rearrangement of the interaction between three fiber molecules and the adaptor/nozzle, leading to the subsequent opening of the DNA channel by releasing restrictions of valves and gates for DNA injection (Fig. 5).

In addition to previously identified valve and gates, the extra valve (valve S) and gate (gate 1S) observed in the structure of mature P-SCSP1u may contribute to securing DNA from escaping through the electrostatic force (Fig. 5). To inject the highly condensed DNA from the icosahedral capsid, the portal-tail complex needs to undergo conformational changes to open the gated central channel. Together with previous studies of T7, the high-resolution structure of mature P-SCSP1u, particularly the portal-tail complex, enabled us to propose a model to understand the DNA gating system at the initiation stage of infection for P-SCSP1u and other T7-like viruses (Fig. 5). Upon the fiber attaching to the host cell, the conformational changes in the tail complex, triggered by the connection between the N-terminal of fiber and portal/adaptor, would facilitate the release of DNA. On one hand, the positively charged valve S may rotate to release DNA from traction in a way like the observation in T7 (Fig. 5). On the other hand, the conformational changes of the nozzle would allow the opening of the gates in the portal for DNA injection (Fig. 5). With these two major restrictions released, the DNA of P-SCSP1u could pass through the portal-tail complex and be injected into the host cell. Notably, this hypothesis is waiting for further verification by determining the high-resolution and intact structure of the tail complex of P-SCSP1u in the DNA-injected state.

## Methods

### Preparation of high-titer phage particles

Ten liters of the host *Prochlorococcus* MED4 cells were grown to mid-exponential phase in Pro99 medium with continuous aeration under a 14 h light (60 µmol photons m$^{-2}$ s$^{-1}$) and 10 h dark cycle at 23 °C. The phage P-SCSP1u with a multiplicity of infection of 0.1 was added to the culture. After complete cell lysis, the lysate was treated with 1 µg/mL DNase I at room temperature for 1 h and then filtered through a 0.2 µm membrane to remove cellular debris. The phage particles were precipitated with 0.5 M NaCl and polyethylene glycol (PEG) 8000 (10% w/v) overnight at 4 °C. Afterward, phage particles from the PEG pellet (10,000 × *g*, 4 °C, 60 min) were resuspended in SM buffer (50 mM Tris, pH 7.5, 100 mM NaCl, 10 mM MgSO$_4$) and purified with cesium chloride (1.5 g/mL in SM buffer) gradient ultracentrifugation (200,000 × *g*, 4 °C, 24 h). The opalescent phage band was collected and dialyzed against SM buffer at 4 °C to remove cesium chloride.

### Negative staining sample preparation and imaging

Before the cryo-EM investigation, the quality of the concentrated phage particles was checked by negative staining transmission electron microscopy. Briefly, 3 µL of phage particles was loaded onto a formvar, carbon-coated copper grid and incubated for 1 min. The phage particles were negatively stained with 2% (w/v) uranyl acetate for 30 s. Excess stain was removed with filter paper. The air-dried grid was imaged with a Talos L120C electron microscope (Thermo Fisher Scientific) operated at 120 kV with

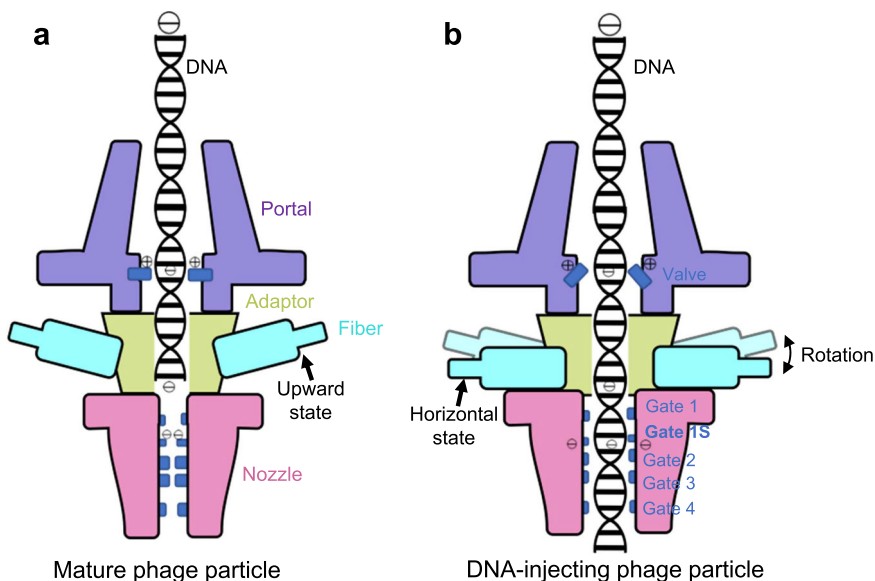

**Fig. 5 | Proposed DNA gating model of P-SCSP1u during the DNA-injecting stage of infection.** The proposed P-SCSP1u portal-tail complexes in the mature (**a**) and DNA-injecting (**b**) states are shown in cartoons with components colored differently. Key elements interacting with DNA are indicated. The possible conformational changes are highlighted and noted.

a 4k × 4k Ceta 16 M camera. A magnification of 57,000×, corresponding to a pixel size of 2.49 Å on the specimen, and a defocus around −1.5 µm were used for recording.

## Cryo-EM sample preparation and data collection
A drop of 3.5 µL purified P-SCSP1u sample was loaded onto a glow-discharged holey carbon grid (Quantifoil R1.2/1.3, 300 mesh Cu). The grids were blotted once for 6 s at 100% humanity and 22 °C and plunged into the liquid ethane using a Mark IV Vitrobot (Thermo Fisher Scientific). The cryo-grids were transferred to Titan Krios G3i electron microscope (Thermo Fisher Scientific) equipped with a high-brightness field emission gun operated at 300 kV for imaging with a K3 summit direct electron detector (Gatan). The slit width of the Gatan Imaging Filter (GIF) Bio Quantum was set to 20 eV for automatic data collection using EPU in super-resolution mode at a calibrated magnification of 53,000× (1.70 Å/pix physical pixel size). The dose rate, total dose, and defocus range used for data collection are summarized in Supplementary Table 1.

## Image processing
MotionCor2 was used to motion-correct 40-frame movie stacks[44]. The motion-corrected micrographs without dose-weighting were used for contrast transfer function (CTF) estimation with GCTF[45]. The motion-corrected micrographs with dose-weighting were used for all other image processing. A total of 293,045 mature P-SCSP1u particles were auto-picked using Gautomatch (https://www.mrclmb.cam.ac.uk/kzhang/Gautomatch/) and extracted in RELION[46]. After several rounds of 2D classifications, 70,233 virion particles with high quality were selected for further processing. An initial map of phage capsid generated by cryoSPARC was used for 3D auto-refine to align these particles with I3 symmetry (Supplementary Fig. 1)[47].

To further improve the resolution of the capsid, sub-particle and symmetry expansion methods[33] were further applied to obtain good virion particles. Briefly, the icosahedral symmetry of 70,233 virion particles was expanded based on 12 vertices of I3 symmetry capsid in RELION[46]. The sub-particles were recentered and re-extracted with a smaller box size, with 842,724 sub-particles generated in total. 3D classification without alignment was performed to discard the low-quality particles, and one class with 732,863 sub-particles was selected for the final 3D refinement. After several rounds of 3D auto-refine, CTF refinement, and post-processing in RELION, the capsid structure was solved at 3.2 Å with C5 symmetry applied.

For the asymmetric reconstruction of the tail machine, the symmetry of 70,233 virion particles was expanded based on 12 vertices of I3 symmetry capsid and C5 symmetry of every vertex to generate 4,213,980 particles in total. All particles were masked to focus on one vertex of the capsid and classified to select only the particles with a tail using masked 3D classification without alignment. Finally, 86,469 selected particles were re-extracted with a larger box size and aligned to a tailed phage map with C5 symmetry at 5.83 Å to locate the positions of the tails. The coordinates of each particle from 3D refinement were modified to center each tail machine of the virus in the box, and the re-extraction was applied to all 86,469 particles with these new coordinates at smaller box sizes. After several rounds of 3D auto-refine and CTF refinement, a map of the P-SCSP1u portal-tail complex at 3.8 Å was yielded with C6 symmetry.

To reveal the mismatching details between the capsid and the portal-tail complex, 3D classification and 3D refinement without symmetry were applied to particles. Based on the coordinates of the portal-tail complex after symmetric refinement, the 86,469 particles with 256 pixels box size were recentered and re-extracted in RELION to focus on the capsid-portal-tail binding part. The particles were transferred into cryoSPARC and symmetry-expanded with the C6 symmetry of the tail. The capsid part of the particles was masked and classified by 3D classification without alignment. Five major classes of particles were generated in output because of the C5 symmetry of the capsid, and the clearest class, with 44,952 particles, was chosen for further reconstruction. Finally, the map of the mismatched capsid-portal-tail complex was solved at 4.6 Å without symmetry by non-uniform refinement. The resolution was estimated using the Gold Standard Fourier Shell Correlation 0.143 criterion.

## Structural comparison
The published structural models of capsid asymmetric units were downloaded from the PDB website for the DALI structural comparison analysis[42]. The C-alpha backbone traces of P-SSP7 and Syn5 were input into the PD2 ca2main web server[48] to reconstruct the full backbone based on the alpha carbon, which could satisfy the format requirement of the DALI server. Finally, the structural similarities of seven chains of all capsids were demonstrated via Z-scores using the all-against-all

structure comparison of the DALI server[42] according to the coordinates of alpha carbon, which avoided the interference of extra atoms from PD2 ca2main.

## Model building
The initial atomic models of capsid protein (gp19) were predicted with the SWISS-MODEL[49], and other initial models of the components in the tail-portal complex (portal gp16, adaptor gp22, fiber gp28, and nozzle gp23) were generated by AlphaFOLD[50]. All models were checked and refined in real space with Phenix[51] and subsequently adjusted manually in Coot[52]. The asymmetric unit atomic model of the P-SCSP1u capsid protein, consisting of one hexon and one subunit of penton, was finally built based on the capsid density map. The adaptor, N-terminal fiber, nozzle, and portal are refined against the density map of the P-SCSP1u portal-tail complex. Chimera and Chimera X were used for structural analysis and comparison[53,54]. Hole 2.0 was used to calculate the radius of the DNA channel in the portal-tail complex[55]. All structure figures were prepared using Chimera X[54]. The detailed information on cryo-EM data collection, processing, and refinement is summarized in Supplementary Table 1.

## Reporting summary
Further information on research design is available in the Nature Portfolio Reporting Summary linked to this article.

## Data availability
Cryo-EM density maps of P-SCSP1u cyanophage have been deposited in the Electron Microscopy Data Bank under accession code EMD-35174 (capsid) and EMD-35175 (portal-tail complex). Atomic coordinates have been deposited in the Protein Data Bank under accession codes 8I4L (capsid) and 8I4M (portal-tail complex). Source data are provided with this paper.

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

## Acknowledgements

We thank Yanxiang Cui for advice on data processing. All cryo-EM data were collected at the Biological Cryo-EM Center at HKUST, generously supported by a donation from the Lo Kwee Seong Foundation. This project is supported by grants of S.D. from Hong Kong Research Grants Council (ECS26101919, GRF16103321, GRF16102822, C7009-20GF, C6001-21EF), Guangdong Basic and Applied Basic Research Foundation (2021A1515012460), Shenzhen Special Fund for Local Science and Technology Development Guided by Central Government (2021Szvup140) and HKUST start-up and initiation grants. The work described in this paper was partially supported by grants of Q.Z. from the Research Grants Council of the Hong Kong Special Administrative Region, China (Project No. HKUST C6012-22GF), Hetao Shenzhen-Hong Kong Science and Technology Innovation Cooperation Zone (HZQB-KCZYB-2020083), and SML99147-42080013/99138-42020015. L.C. acknowledges support from the Senior User Project of RV KEXUE (KEX-UE2020G09). CORE is a joint research center for ocean research between QNLM and HKUST.

## Author contributions

Q.Z. and S.D. supervised the project. L.C. and S.X. purified the P-SCSP1u cyanophage; W.Z. and S.D. prepared cryo-sample and collected cryo-EM datasets; H.L. and S.D. processed images and built the atomic model. L.C., H.L., Q.Z., and S.D. analyzed the data, prepared the figures, and wrote the manuscript.

## Competing interests

The authors declare no competing interests.
