## [Peer Review File · Nature Communications]

Reviewers' Comments:

Reviewer #1:

Remarks to the Author:

In this manuscript, Cai et al. report their cryoEM structural studies of a cyanobacterial phage P-SCSP1u that is closely related to the better-known phage T7. They solved not only structure of the icosahedral capsid, but also structures of the portal and tail complex with more advanced data processing techniques including the sub-particle and symmetry expansion methods. Although it is becoming routine to solve structures of asymmetric components in icosahedral viruses, particularly for feature-rich targets like this one, still it requires a significant amount of effort. Unfortunately, compared to the admirable achievement in structure determination, the presentation of the structure in the current version of the manuscript is shallow and not much insightful. The most significant discovery is the extra valve and gate in the tail and portal region compared to other T7-like viruses, which provides interesting insights into the mechanism of DNA gating in tailed phages. I would like to suggest that the authors should expand their structure analysis and description in the following aspects to make this paper more interesting and significant.

1. For the capsid part, the authors only described the subunit structure of the major capsid protein, but did not provide any detail about their interactions. How is the capsid assembled to withstand the high pressure of the dsDNA genome without reinforcement by cement proteins or forming covalent bonds like those in HK97? Does the N-arm form lasso-like structures as those described in herpesvirus capsids?
2. There is a symmetry mismatch between the 5-fold penton vertex and the 12- or 6-fold portal-tail complex. How is the symmetry-mismatched portal-tail complex accommodated at the special penton vertex? Is there any structural difference of the capsid protein at this vertex compared to other penton vertices?
3. Lines 183-196: The authors made the description of interactions among the different portal-tail components extremely over-simplified and vague. It is hard to believe that such a stable structure that can withstand tens of atmospheres of pressure exerted by the internal dsDNA genome is just organized by such few hydrogen bonds as described in the text. A more serious analysis of protein-protein interface interactions by programs such as PISA should be provided and described. Minor technical issues: arginine (R766) and glutamic acid (E20) should be forming salt bridge instead of hydrogen bond; how can the hydrophobic isoleucine residues (I321, I145) form hydrogen bonds with the others?
4. The authors propose to use structural comparison of the capsid proteins to (re)-define the phylogeny of the closely related cyanophages. I do not think this kind of analysis with structural proteins (in this case, the capsid proteins) is as reliable as sequence comparison of nonstructural proteins, such as the polymerases, used in most phylogeny analysis. The capsid proteins may have to adapt different conformations in the extended loop regions, or have extra insertions (which happens to be the case in this paper) or deletions of a few amino acids to build a different size of capsid and to accommodate different sizes of dsDNA genome. Such structural changes would generate a drastically increased RMSD that not necessarily reflecting the true revolution distance between the two viruses. Therefore, I would suggest that this part should be removed.

Other minor points:

5. The components described in the text-- "portal", "adaptor", "nozzle" etc. should be labeled on Figures. 1C, 1E to facilitate apprehension of the structure.
6. Lines 128-131. The description seems to be wrong. Should be "one central penton and five surrounding hexons"? In addition, it is usually called icosahedral "vertex", not "apex".
7. In Fig. 2C, two copies of the Portal, Adaptor, or Nozzle subunits across the central channel should be displayed to show the relative positions of each domain, and to help the reader comprehend the description like "the valve ... points perpendicularly to the channel" (lines 158-161).

Reviewer #2:

Remarks to the Author:

This manuscript describes the CryoEM single particle structure of a marine bacteriophage. The capsid and portal complex were determined to 3.2 and 3.8 Angstrom (Å) resolution, respectively. The authors draw conclusions about the packaging and release of the DNA from the capsid.

I have a few questions. Most of these concern the capsid protein (gp19) structure and comparisons with other cyanophages (Syn5 and P-SSP7).

1. In examining the PDB Validation Report, the geometry (bond-lengths and angles) of the model was quite reasonable, so no great liberties were taken to force the model to conform to the map, which is good. However, the atom inclusion and Q-scores (sections 9.2-9.5) both show a progressive degradation of the map-model fit. This falls off as the distance from the "anchored" location of the 5-fold vertex. The falloff, especially in the atom-inclusion metric, appears to be almost linear with distance from the 5-fold. I suspect that the Å/pixel of the map is slightly off. This is not uncommon, and usually shows up at this stage of the structure determination. Since the clash-score is not bad, the Å/pixel may have been too large, rather than too small, because the atoms are not squeezed together. It must be emphasized that the image processing that created the map could be perfectly valid and probably doesn't need to be revisited (except for very minor effects on the CTF determination, which are a second-order effect), but the Å/pixel should be "refined" so that models with good geometry fit homogeneously into the rescaled map.

2. The discussion of the comparisons with capsid proteins from other cyanophages was good, and very interesting, given that this capsid is unusually small. However, the differences might be brought into context with more comparisons. I suggest comparing (once #1 is addressed) all seven of the chains in this structure with each other, since they should have the least RMSD and of course have identical sequences. This will contextualize the differences with the other cyanophages. I'd also, on the high end, include comparisons with other phages, gp5 from phage P22, for instance, which should be even more different.

The PDB Validation Report was critical in evaluating this manuscript.

Response to Reviewer comments

We thank the reviewers for their constructive and detailed comments. We have thoroughly revised the manuscript to address all the points raised by the reviewers. As a result, the quality and accessibility of the manuscript have been significantly improved.

Reviewer #1 (Remarks to the Author):

In this manuscript, Cai et al. report their cryo-EM structural studies of a cyanobacterial phage P-SCSP1u that is closely related to the better-known phage T7. They solved not only structure of the icosahedral capsid, but also structures of the portal and tail complex with more advanced data processing techniques including the sub-particle and symmetry expansion methods. Although it is becoming routine to solve structures of asymmetric components in icosahedral viruses, particularly for feature-rich targets like this one, still it requires a significant amount of effort. Unfortunately, compared to the admirable achievement in structure determination, the presentation of the structure in the current version of the manuscript is shallow and not much insightful. The most significant discovery is the extra valve and gate in the tail and portal region compared to other T7-like viruses, which provides interesting insights into the mechanism of DNA gating in tailed phages. I would like to suggest that the authors should expand their structure analysis and description in the following aspects to make this paper more interesting and significant.

Response: Thanks for taking the time and effort to help us improve the manuscript. We appreciate the positive feedback and the detailed suggestions. In the revised manuscript, we have performed extensive structural analysis to provide comprehensive protein-protein interactions for the assembly of the virion, including the subunit interactions within/between capsomeres in the capsid, the interactions of protein components in the portal-tail complex, as well as the symmetry mismatch between the five-fold capsid vertex and the 12-fold portal-adaptor complex. Please see the point-by-point responses below and the revised manuscript for more details.

1. For the capsid part, the authors only described the subunit structure of the major capsid protein, but did not provide any detail about their interactions. How is the capsid assembled to withstand the high pressure of the dsDNA genome without reinforcement by cement proteins or forming covalent bonds like those in HK97?

Does the N-arm form lasso-like structures as those described in herpesvirus capsids?

Response: Thanks for the constructive suggestion. As suggested, we have identified the interactions of both intra- and inter-penton/-hexon capsomeres to reveal how the capsid protein interacts with other subunits for the stabilization of the capsid (Figure 1d, Supplementary Figure 3). The pentons and hexons are assembled by similar intra-capsomeric interactions, with A-domains constituting the core of capsomeres and the “head-to-tail” interactions between adjacent E-loops, N-arms, and P-domains, forming a circulating interaction mode to stabilize the capsomer (Figure 1d, Supplementary Figure 3b, 3c). The inter-capsomeric interactions are mediated by mutual penetration of the E-loops and P-domains of neighboring capsomeres and further reinforced by the

extended N-arms interacting with multiple subunits to fasten the adjacent capsomers (Supplementary Figure 3d–f). PISA analyses further revealed substantial hydrogen bonds formed among subunits to provide strong support for the assembly of the capsid (Supplementary Figure 3b–f). Specifically, the N-arm, E-loop, and P-domain of the adjacent subunits from three capsomers interact with each other with more than 30 hydrogen bonds to tightly connect the capsomers. Overall, the capsid of P-SCSP1u is assembled via non-covalent interactions to maintain the high internal pressure for DNA packaging, which is different from that of HK97. The long-distance reinforcing function of N-arms in P-SCSP1u is more similar to the N-lasso structure in herpesviruses. The information has been added in the revised manuscript (Lines 146–167).

2. There is a symmetry mismatch between the 5-fold penton vertex and the 12- or 6-fold portal-tail complex. How is the symmetry-mismatched portal-tail complex accommodated at the special penton vertex? Is there any structural difference of the capsid protein at this vertex compared to other penton vertices?

Response: Thanks for raising the point. To reveal more details about the symmetry mismatch between the five-fold capsid vertex and the 12-fold portal-tail complex, we reconstructed the cryo-EM density map of the portal-tail complex without symmetry at 4.6 Å resolution (Supplementary Figures 1, 4). The result showed that the portal-tail complex replaced one penton of the icosahedral capsid and clipped by five hexons (namely mismatched hexons), in a manner analogous to the pentons in other vertices. Structural comparison of capsid proteins between mismatched hexons and other vertices revealed only slight conformational changes in N-arm and P-domain regions of the capsid proteins (Supplementary Figure 5a). However, various interactions occurred between the capsid and the portal-adaptor in the region (namely portal vertex) (Supplementary Figure 4c, 4d). The N-arm, P-domain, and E-loop of the capsid subunits from five mismatched hexons form a ring-like structure and are clipped by the circular groove of portals and adaptors. The large wing regions of the dodecameric portal form a wide surface for the capsid to bind from inside, while the embracing helices of the adaptor provide a smaller surface to stabilize the central part of the capsid shell (Supplementary Figure 4d, 4e). The information has been added in the revised manuscript (Lines 204–221).

3. Lines 183-196: The authors made the description of interactions among the different portal-tail components extremely over-simplified and vague. It is hard to believe that such a stable structure that can withstand tens of atmospheres of pressure exerted by the internal dsDNA genome is just organized by such few hydrogen bonds as described in the text. A more serious analysis of protein-protein interface interactions by programs such as PISA should be provided and described. Minor technical issues: arginine (R766) and glutamic acid (E20) should be forming salt bridge instead of hydrogen bond; how can the hydrophobic isoleucine residues (I321, I145) form hydrogen bonds with the others?

Response: Thanks for your suggestions. As suggested, we have added the PISA analysis in the revised manuscript. According to the PISA result, one portal subunit formed four hydrogen bonds and one salt bridge with two adaptor subunits, resulting in a total of 48 hydrogen bonds and 12 salt bridges formed between the dodecameric portal protein and dodecameric adaptor protein. Similarly, the dodecameric adaptor protein formed 12 hydrogen bonds and 6 salt bridges with the hexameric nozzle protein. The results showed that 10 residues of one portal subunit (P1) and 15 residues of the adaptor subunit (A2) produced an interface area of around 500 Å² and 22 residues of P2 and 24 residues of A2 formed around 700 Å² interface area. An approximate 500 Å² area was formed by 16 residues of N1 and 18 residues of A3. Another 14 residues of N1 interacted with 15 residues of another adaptor subunit (A2), forming a 420 Å² interface. Collectively, the dodecameric portal, the dodecameric adaptor, and the hexameric nozzle in the portal-tail complex are intimately connected to each other with 60 hydrogen bonds, providing a stable platform for DNA ejection. The information has been added in the revised manuscript (Lines 229–245).

For the interaction between R766 and E20, the hydrogen bonds were formed by the side chain of R766 and the main chain of E20. The salt bridge was formed between R766 and E19. The description of R766-E19 and R309-E50 has been revised as 'salt bridge' and indicated as red dashed lines in Figure 3b-c. The hydrogen bonds of I145 were removed in the revised manuscript. The hydrogen bond between I321 and H176 was formed by the main chain and preserved according to the PISA analysis.

4. The authors propose to use structural comparison of the capsid proteins to (re)-define the phylogeny of the closely related cyanophages. I do not think this kind of analysis with structural proteins (in this case, the capsid proteins) is as reliable as sequence comparison of nonstructural proteins, such as the polymerases, used in most phylogeny analysis. The capsid proteins may have to adapt different conformations in the extended loop regions, or have extra insertions (which happens to be the case in this paper) or deletions of a few amino acids to build a different size of capsid and to accommodate different sizes of dsDNA genome. Such structural changes would generate a drastically increased RMSD that not necessarily reflecting the true evolution distance between the two viruses. Therefore, I would suggest that this part should be removed.

Response: Thanks for your suggestions. We agree with the reviewer that capsid proteins might adopt conformational adjustments to accommodate different radii of the capsid shell. Similarly, subunits in different positions (e.g., subunits in pentons and hexons) would show conformational alterations. To evaluate the potential influence caused by the intrinsic flexibility of capsid proteins, seven subunits in the viral asymmetric unit of different phages (including T7, T7-like cyanophages, HK97, and *Salmonella* phage P22), as suggested by Reviewer #2, are used for pairwise structure comparison. Subunits were structurally aligned against each other to determine their structural similarities via the widely used DALI Z-scores analysis. The hierarchical clustering analysis based on structural similarity revealed that capsid proteins from the same phage are clearly clustered together (Supplementary Figure 6), indicating that structural

comparisons of capsid proteins among different phages are not obscured by the conformational shifts of identical capsid proteins during assembly. Furthermore, consistent with our previous result, the capsid protein subunits of P-SCSP1u show higher structural similarities with those of T7, even though gp19 (capsid protein) of P-SCSP1u shares slightly higher sequence similarity with gp39 (capsid protein) of Syn5. Thus, proteins with high sequence similarity do not necessarily have high structural similarity. The results of our analysis provide a glimpse into the potential application and development of the structure-based classification of proteins. We proposed that as more structures of proteins are solved, structure-based analysis, in combination with sequence-based analysis, has the increasing potential to optimize the classification of viruses at a holistic level. The information has been added in the revised manuscript (Lines 353–371).

Other minor points

5. The components described in the text-- “portal”, “adaptor”, “nozzle” etc. should be labeled on Figures. 1C, 1E to facilitate apprehension of the structure.

Response: Thanks for your suggestion. The previous Figure 1c has been presented as Figure 2a in the revised manuscript. As suggested, the labels are added in the revised Figure 2a for clarity.

6. Lines 128-131. The description seems to be wrong. Should be “one central penton and five surrounding hexons”? In addition, it is usually called icosahedral “vertex”, not “apex”.

Response: Thanks for your correction. The sentence has been revised as “Each vertex of the P-SCSP1u icosahedral head includes one central penton and five surrounding hexons” (Line 130).

7. In Fig. 2C, two copies of the Portal, Adaptor, or Nozzle subunits across the central channel should be displayed to show the relative positions of each domain, and to help the reader comprehend the description like “the valve ... points perpendicularly to the channel” (lines 158-161).

Response: As suggested, two copies of the Portal, Adaptor, and Nozzle subunits across the central channel are shown in the revised Figure 2b in order to clearly display the relative position and interactions between domains. Domains of each subunit are also labeled in the revised Figure 2d.

Reviewer #2 (Remarks to the Author):

This manuscript describes the CryoEM single particle structure of a marine bacteriophage. The capsid and portal complex were determined to 3.2 and 3.8 Angstrom (Å) resolution, respectively. The authors draw conclusions about the packaging and release of the DNA from the capsid. I have a few questions. Most of these concern the capsid protein (gp19) structure and comparisons with other cyanophages (Syn5 and P-SSP7).

Response: Thanks for taking the time and effort to help us improve the manuscript. We appreciate the positive feedback and the professional comments. According to your suggestion, we have double-checked the PDB Validation Report. Please see the point-by-point responses below and the revised manuscript for more details.

1. In examining the PDB Validation Report, the geometry (bond-lengths and angles) of the model was quite reasonable, so no great liberties were taken to force the model to conform to the map, which is good. However, the atom inclusion and Q-scores (sections 9.2-9.5) both show a progressive degradation of the map-model fit. This falls off as the distance from the “anchored” location of the 5-fold vertex. The falloff, especially in the atom-inclusion metric, appears to be almost linear with distance from the 5-fold. I suspect that the Å/pixel of the map is slightly off. This is not uncommon, and usually shows up at this stage of the structure determination. Since the clash-score is not bad, the Å/pixel may have been too large, rather than too small, because the atoms are not squeezed together. It must be emphasized that the image processing that created the map could be perfectly valid and probably doesn't need to be revisited (except for very minor effects on the CTF determination, which are a second-order effect), but the Å/pixel should be “refined” so that models with good geometry fit homogeneously into the rescaled map.

Response: Thanks for your careful inspection and professional advice. The Q-scores decrease as the increase of the distance from the “anchored” location, which is also the center of the reconstructed map. To improve the resolution of the capsid protein, we performed sub-particle and symmetry expansion to align penton in the center for re-extraction of particles during the data processing. For the final reconstruction, C5 symmetry is applied. In this way, the resolution of the cryo-EM density map closing to the edge of the box shows a lower resolution than that in the center of the box (Figure R1), which is common for sing-particle analysis. Furthermore, the resolution drop in the box edge region may partially result from structural heterogeneity of the capsid proteins at different positions, which is supported by slight structural differences of the capsid protein in the asymmetry unit (Supplementary Figures 5a, 6). The resolution decreasing would cause the falloff of the Q scores, as well as the atom inclusion in the PDB Validation Report.

To exclude the possibility that the pixel size is slightly off, we manually assigned pixel sizes around 0.85Å for the cryo-EM density map, and calculated model-map cross-correlation (CC) for each assigned pixel size. The results showed that model-map CC decreased when assigned pixel size off from 0.85Å (Figure R2), which further demonstrated the accuracy of the pixel size for our map. Moreover, the structure of the

major capsid protein was built against density in the center of the box, which possesses the highest map quality and Q-score. Therefore, our structural interpretation and main conclusion will not be affected.

Figure R1 Local resolution of the cryo-EM density map of capsid penton. The resolution of the density map closing to the edge of the box dropped slightly.

Figure R2 Verification of pixel size for cryo-EM density map. Calculated cross-correlation (CC) values between the model and cryo-EM density map with assigned pixel size are indicated. The largest CC value is observed when the pixel size equals to 0.85 Å.

2. The discussion of the comparisons with capsid proteins from other cyanophages was good, and very interesting, given that this capsid is unusually small. However, the differences might be brought into context with more comparisons. I suggest comparing (once #1 is addressed) all seven of the chains in this structure with each other, since they should have the least RMSD and, of course, have identical sequences. This will contextualize the differences with the other cyanophages. I'd also, on the high end, include comparisons with other phages, gp5 from phage P22, for instance, which should be even more different.

Response: Thanks for your positive feedback and detailed suggestions. As suggested, seven protein subunits in the viral asymmetric unit of different phages (including T7, T7-like cyanophages, HK97, and *Salmonella* phage P22) are all used for pairwise structure comparison. Subunits were structurally aligned against each other to determine their structural similarities via the widely used DALI Z-scores analysis. The hierarchical clustering analysis based on structural similarity showed that capsid proteins from the same phage are clearly clustered together (Supplementary Figure 6). Consistent with our previous result, the capsid protein subunits of P-SCSP1u have higher structural similarities with those of T7, even though gp19 (capsid protein) of P-SCSP1u shares slightly higher sequence similarity with gp39 (capsid protein) of Syn5. Thus, proteins with high sequence similarity do not necessarily have high structural similarity. We proposed that as more structures of proteins are solved, structure-based analysis, which allows for the detection and quantification of architectural properties between proteins, has the increasing potential to draw evolutionary links between proteins and optimize the classification of viruses in future studies. The information has been added in the revised manuscript (Lines 353–371).

The PDB Validation Report was critical in evaluating this manuscript.

Response: We totally agree with the reviewer that the PDB Validation Report is important for structural studies. As mentioned, we carefully verified the accuracy of the pixel size and further checked the models we built in this study to ensure the high quality of structural interpretation and analyses in this manuscript.

Reviewers' Comments:

Reviewer #1:

Remarks to the Author:

The authors have addressed all my concerns raised from the previous version of the manuscript.

Reviewer #2:

Remarks to the Author:

1. I thank the authors for investigating the effect of the pixel size on the quality of the cross-correlation fit of the model to the map. From the description of Figure R2, it is unclear if they did a cycle of real-space refinement (in phenix, for instance) of their models, for each of the rescaled maps, before doing the cross-correlation, which is what I was suggesting, or whether they only checked the fit of their static (unchanged) model into maps that were rescaled, perhaps after a rigid-body fit. I expect that the models should have to change slightly for each of the different rescaled maps and that there would be an optimum refinement into whichever map had the correct pixel size. I apologize if I was not clear.

The fact that the local resolution of the map varies only slightly (from 2.7 to 3.3Å) in figure R1 means that the map itself is very good and I agree with them that the structural interpretation and main conclusion will not be affected.

2. For the structure/sequence comparison, I thank the authors for the new Supplementary Figures 5 and 6. Both of them illustrate the structural similarities among the capsid proteins, and that the penton subunit has a higher diversity in structure within and between the capsid families.

Response to Reviewer comments

Reviewer #1 (Remarks to the Author):

The authors have addressed all my concerns raised from the previous version of the manuscript.

Response: Thanks for taking the time and effort to help us improve the manuscript.

Reviewer #2 (Remarks to the Author):

1. I thank the authors for investigating the effect of the pixel size on the quality of the cross-correlation fit of the model to the map. From the description of Figure R2, it is unclear if they did a cycle of real-space refinement (in phenix, for instance) of their models, for each of the rescaled maps, before doing the cross-correlation, which is what I was suggesting, or whether they only checked the fit of their static (unchanged) model into maps that were rescaled, perhaps after a rigid-body fit. I expect that the models should have to change slightly for each of the different rescaled maps and that there would be an optimum refinement into whichever map had the correct pixel size. I apologize if I was not clear.

The fact that the local resolution of the map varies only slightly (from 2.7 to 3.3Å) in figure R1 means that the map itself is very good and I agree with them that the structural interpretation and main conclusion will not be affected.

Response: We appreciate the reviewer's positive feedback and detailed suggestions. We apologize for the confusion regarding Figure R2. Initially, we used the same model, without real-space refinement, for fitting into individual rescaled cryo-EM density maps in Chimera to calculate the CC value. However, following the reviewer's instructions, we have now re-calculated the CC value to verify the pixel size. In this iteration, we performed phenix.real_space_refine to generate the corresponding model for each rescaled map. Subsequently, we calculated the CC value between each rescaled map and its respective model. As depicted in Figure R3, the accurate pixel size is found to be very close to 0.85 Å. Additionally, we found the refinement statistics (Table R1) of the models, refined against the rescaled cryo-EM density map, were slightly worse compared to the original model using a pixel size of 0.85 Å. This observation further confirms that 0.85 Å is a more accurate pixel size for our map.

Figure R3 Verification of pixel size for cryo-EM density map. The plot shows the cross-correlation (CC) value, calculated between the rescaled cryo-EM density map and the corresponding refined model in real space, for each assigned pixel size. To account for data fluctuates, a polynomial fitting analysis was applied. The polynomial trendline (magenta) represents the overall relationship observed. The peak position of the CC value indicates that the accurate pixel size is very close to 0.85 Å.

Table R1 Refinement statistics of models against the corresponding map with the assigned pixel size.

pixel size (Å)	Ramachandran plot (%)			Rotamer outliers (%)	C-β outliers (%)	Bonds (RMSD)	
	Favored	Allowed	Outliers			Length (Å)	Angles (°)
0.82	93.58	6.42	0	0.27	0	0.006	0.916
0.83	95.12	4.88	0	0.44	0	0.004	0.611
0.84	94.64	5.36	0	0.27	0	0.003	0.55
0.845	93.23	6.73	0.04	0.6	0	0.003	0.585
0.85	95.38	4.62	0	0.16	0	0.002	0.498
0.855	93.32	6.68	0	0.38	0	0.003	0.565
0.86	93.98	6.02	0	0.11	0	0.007	0.85
0.87	92.92	7.08	0	0.38	0	0.007	0.885
0.88	94.81	5.19	0	0.49	0	0.002	0.554
0.89	93.71	6.29	0	0.44	0	0.003	0.646

2. For the structure/sequence comparison, I thank the authors for the new Supplementary Figures 5 and 6. Both of them illustrate the structural similarities among the capsid proteins, and that the penton subunit has a higher diversity in structure within and between the capsid families.

Response: Thanks for taking the time and effort to help us improve the manuscript.